# Renal Denervation Helps Preserve the Ejection Fraction by Preserving Endocardial-Endothelial Function during Heart Failure

**DOI:** 10.3390/ijms24087302

**Published:** 2023-04-15

**Authors:** Sathnur Pushpakumar, Mahavir Singh, Yuting Zheng, Oluwaseun E. Akinterinwa, Sri Prakash L. Mokshagundam, Utpal Sen, Dinesh K. Kalra, Suresh C. Tyagi

**Affiliations:** 1Department of Physiology, School of Medicine, University of Louisville, Louisville, KY 40202, USA; 2Division of Endocrinology, Metabolism and Diabetes and Robley Rex VA Medical Center, University of Louisville School of Medicine, Louisville, KY 40202, USA; 3Division of Cardiovascular Medicine, University of Louisville School of Medicine, Louisville, KY 40202, USA

**Keywords:** antioxidant molecules, cardiac health, mechanism of heart failure, cardiac activity

## Abstract

Renal denervation (RDN) protects against hypertension, hypertrophy, and heart failure (HF); however, it is not clear whether RDN preserves ejection fraction (EF) during heart failure (HFpEF). To test this hypothesis, we simulated a chronic congestive cardiopulmonary heart failure (CHF) phenotype by creating an aorta-vena cava fistula (AVF) in the C57BL/6J wild type (WT) mice. Briefly, there are four ways to create an experimental CHF: (1) myocardial infarction (MI), which is basically ligating the coronary artery by instrumenting and injuring the heart; (2) trans-aortic constriction (TAC) method, which mimics the systematic hypertension, but again constricts the aorta on top of the heart and, in fact, exposes the heart; (3) acquired CHF condition, promoted by dietary factors, diabetes, salt, diet, etc., but is multifactorial in nature; and finally, (4) the AVF, which remains the only one wherein AVF is created ~1 cm below the kidneys in which the aorta and vena cava share the common middle-wall. By creating the AVF fistula, the red blood contents enter the vena cava without an injury to the cardiac tissue. This model mimics or simulates the CHF phenotype, for example, during aging wherein with advancing age, the preload volume keeps increasing beyond the level that the aging heart can pump out due to the weakened cardiac myocytes. Furthermore, this procedure also involves the right ventricle to lung to left ventricle flow, thus creating an ideal condition for congestion. The heart in AVF transitions from preserved to reduced EF (i.e., HFpEF to HFrEF). In fact, there are more models of volume overload, such as the pacing-induced and mitral valve regurgitation, but these are also injurious models in nature. Our laboratory is one of the first laboratories to create and study the AVF phenotype in the animals. The RDN was created by treating the cleaned bilateral renal artery. After 6 weeks, blood, heart, and renal samples were analyzed for exosome, cardiac regeneration markers, and the renal cortex proteinases. Cardiac function was analyzed by echocardiogram (ECHO) procedure. The fibrosis was analyzed with a trichrome staining method. The results suggested that there was a robust increase in the exosomes’ level in AVF blood, suggesting a compensatory systemic response during AVF-CHF. During AVF, there was no change in the cardiac eNOS, Wnt1, or β-catenin; however, during RDN, there were robust increases in the levels of eNOS, Wnt1, and β-catenin compared to the sham group. As expected in HFpEF, there was perivascular fibrosis, hypertrophy, and pEF. Interestingly, increased levels of eNOS suggested that despite fibrosis, the NO generation was higher and that it most likely contributed to pEF during HF. The RDN intervention revealed an increase in renal cortical caspase 8 and a decrease in caspase 9. Since caspase 8 is protective and caspase 9 is apoptotic, we suggest that RDN protects against the renal stress and apoptosis. It should be noted that others have demonstrated a role of vascular endothelium in preserving the ejection by cell therapy intervention. In the light of foregoing evidence, our findings also suggest that RDN is cardioprotective during HFpEF via preservation of the eNOS and accompanied endocardial-endothelial function.

## 1. Introduction

After initial publications mentioning the HFpEF phenotype, starting in the 1980s, the studies on HFpEF were mainly of case series types or the comparative clinical descriptions from patients with HF. These studies revealed significant variability in the HFpEF prevalence. However, a report from the Helsinki Study in 1987 that had sampling from 501 people helped demonstrate the preserved left ventricle systolic function in more than 50% of the prevalent cases. Subsequently, numerous cohorts also reported on HFpEF prevalence in the patients including the Framingham Heart Study and the Strong Heart Study and the Cardiovascular Health Study. Almost every one of them confirmed HFpEF phenotype in large proportions of HF cases. These observations were also concordant with that of the data from the hospitalized patients and from large HF registries. Interestingly, the HFpEF confirmation in those studies was based on the combination of the HF—assessed based on clinical symptoms, signs, and the left ventricle ejection fraction (LVEF) threshold values without considering the data from atrial or ventricular structure or diastolic function [1,2,3,4,5,6,7,8,9,10].

Clinical diagnosis and management of heart failure (HF) are currently linked to ejection fraction (EF). HF with preserved ejection fraction (HFpEF) is now considered a distinct disease from HF with reduced ejection fraction (HFrEF). Although the morbidity and mortality rates of HFpEF and HFrEF are comparable, therapeutic options for HFpEF patients remain elusive. The HFpEF was initially thought to be due to diastolic dysfunction but is now recognized as a complex interplay of impairment in cardiac reserve, as well as altered renal metabolism, vascular, pulmonary, and renal nerve activation [11,12,13,14]. Thus, there is a genuine interest in studying the association of HFpEF with extra-cardiac features, including disorders of renal feedback and associated metabolic regulation. Currently, there is no effective cure or therapy for HFpEF. Hence, the incidence and prevalence of HFpEF continue to increase [12,15,16,17,18,19,20]. In that very context, finding potential novel targets is an important unmet need.

The reciprocal contribution by other organs to redistribute blood volume has been observed during myocardial infarction during hind-limb remote ischemic condition [21]. The contribution of skeletal muscle myokines, sympathetic inactivation, and bone marrow induction, along with exercise and muscle myokine/musclin HF, has been suggested; however, the reciprocal contribution in HFpEF to HFpEF is unclear [22,23,24,25,26]. Our recent study shows that the remote hind-limb ischemia releases exosome containing musclin that contributes to the HFpEF [27]. Joslin et al. have shown interrelation between heart failure with preserved ejection fraction and renal impairment [28]. Renal denervation has been shown to be cardioprotective [29,30,31]. Angiopoietins released from the kidney play a protective role in heart failure [32]. Moreover, congenital heart disease in adults is associated with autosomal dominant polycystic kidney disease [33]. The mitochondrial implication as potential mediator of systemic inflammation, and organ crosstalk in the causation of acute kidney injury has been suggested [34]. During volume overload, the kidneys sense the channels to release extracellular ATP, and thus compensate the systemic organs [35]. However, it is unclear whether volume overload activates the renal sympathetic nerve, helps excrete the exosomes containing erythropoietin (EPO), and increase the regeneration capacity of the heart. Recently, Marban and colleagues have shown that cell therapy preserves ejection fraction (EF) via the vascular endothelium [36]. Here, we show that renal denervation preserves ejection fraction via endocardial endothelium. Erythropoietin is released from kidneys that, in turn, induces bone marrow parenchyma in releasing the regenerating cell population [37,38]. Therefore, it is novel to propose that increased EPO release via the exosomes enhances endothelial nitric oxide synthase (eNOS), and cardiac regeneration by anti-apoptotic effects via caspases.

## 2. Results

To determine whether the chronic volume overload causes increase in exosomes’ release in the blood of C57BL/6J experimental mice, we created AVF and at 16 weeks, measured the blood levels of exosomes by marking heat shock protein 70 (HSP70). The results revealed an increase in Exosomal amount in AVF mice group compared to WT mice. To mitigate AVF induced volume receptors we created renal denervation (RDN). The results suggested that RDN has tendency to increase exosomes in AVF mice compared with that of the RDN mice alone, but it was lower than the AVF group. Similarly, the levels of erythropoietin (EPO) were lower in the exosomes released during RDN intervention. Interestingly, the levels of exosomes secreted in AVF was significantly higher than the WT mice group. These results suggested that volume stress does cause an increase in the overall exosomes released independently to that of the renal denervation (Figure 1).

To determine whether the AVF suppresses the regeneration and RDN procedure improves cardiac regeneration, we measured the levels of Wnt1 and β-catenin. The results demonstrated the increase in Wnt1 and β-catenin expression post RDN procedure in the mice, thus suggesting an improvement of cardiac regeneration by RDN during HFrEF to maintain HFpEF phenotype (Figure 2). Interestingly, the expression levels of eNOS were found to be robust in the renal denervation mice group, thus indicating the involvement in the cardiac endothelial function by RDN during HFrEF (Figure 2). To determine whether RDN causes renal apoptosis and intracellular renal remodeling, we measured the levels of caspase 8 and caspase 9. The results suggested an increase in caspase 8 levels in RDN and AVF with RDN mice groups. These findings revealed that renal denervation causes renal intracellular remodeling and thereby improves cardiac function during HFrEF (Figure 2).

Next, we wanted to determine whether a putative increase in eNOS during RDN procedure will improve the endocardial-endothelial dependent cardiac function; hence, we measured the endocardial ring response (prepared from the cardiac left ventricle (LV) from C57BL/6J WT-control, AVF, RND, and AVF-RND mice groups) to acetylcholine (ACH). The results demonstrated an attenuation in endothelial dependent cardiac function in AVF group of mice; however, the RDN procedure was able to improve the cardiac endothelial function in experimental animals (Figure 3). Further, to investigate whether the RDN intervention maintains HFpEF phenotype rather than the HFrEF status, we decided to measure the cardiac echocardiographic (ECHO) parameters in the mice groups to quantitate the changes in percentage of the ejection fraction (EF). Our results suggested that RDN procedure successfully improved the cardiac EF percentage in the AVF mice group as shown in Figure 4. Finally, to determine whether the RDN procedure also influences changes such as cardiac hypertrophy and fibrosis, we set out to measure the heart/body weight ratio and collagen contents by measuring hydroxy proline. The results suggested that there were strong indications of cardiac hypertrophy and fibrosis in the AVF hearts, but the RDN procedure did not have any significant effect(s) on the cardiac hypertrophy and fibrosis (Figure 5).

## 3. Discussion

HF is a serious medical emergency that encompasses multiple causes. In the clinic, it is usually diagnosed by a group of clinical symptoms and signs. Most of these presentations relate to the fact that the patients’ ventricles operate at relatively higher filling pressure, because this kind of congestive HF pathophysiology coincides with the spectrum of left ventricle ejection fraction (LVEF). It should be emphasized that regardless of the LVEF modus operandi, the clinical recognition of the symptoms and signs of HF are usually associated with major cardiac adverse events, thus affecting the quality and quantity of patients’ lives. The recognition and appreciation that approximately 50% of patients with this disease have almost normal or close to normal LVEF function makes this even more important, with the results stemming from the major clinical trials demonstrating differential responses to the HF therapies based on LVEF profile.

Surprisingly, almost 50% of the patients tend to have distinct symptoms and related signs of HF with markedly normal left ventricular. This led to the recognition of the HF problem with pEF in the past 25 years that has truly spurred various clinical investigations coupled with informative outcome trials [20,39]. Skeletal muscle denervation increases the exosomes (Figure 1) that often contain miRNAs; however, it is unclear whether renal denervation increases the Exosomal output during HFpEF phenotype [40]. In this study, our results suggested that chronic volume overload causes an increase in exosomes that are released in the blood of the experimental mice. The results revealed that there was a significant increase in the Exosomal contents in the AVF mice group in comparison to the C57BL/6J WT mice. Further, it implies that the RDN procedure tends to increase the exosomes in AVF compared with RDN intervention, but, interestingly, it was lower than the AVF procedure alone. The levels of erythropoietin (EPO) were lower in the exosomes released during RDN; however, it is possible that the respective cargo levels such as miRNAs related to hypertrophy and fibrosis are increased as well.

Others have shown an increase in eNOS levels and a concomitant decrease in hypertrophy by renal denervation procedure [41,42]. Our study corroborates and shows that the levels of eNOS increases and, in fact, were found to be robust in renal denervation, thus suggesting its involvement in the cardiac endothelial function by RDN procedure during HFrEF (Figure 2). Regarding whether the RDN process is responsible for renal parenchymal cells’ apoptosis and the renal intracellular remodeling, we demonstrated the role of caspases (caspase 8 and −9). An increase in caspase 8 level in the RDN procedure and AVF, especially with RDN intervention, was notable, as shown in Figure 2, meaning that renal denervation most likely causes renal intracellular remodeling and helps improve the cardiac function during HFrEF phenotype (Figure 2). In addition, the RDN procedure might also be participating in promoting the cardiac tissue regeneration. It has been also noted that RDN truly has good effects on cardiac tissue remodeling, including beneficial function in resistant hypertension. When cardiac angiogenesis during prolonged pressure overload was studied, it was observed that RDN had a benefit on cardiac angiogenesis, especially during the sustained pressure overload phenotype that essentially involved the regulation of both the vascular endothelial growth factor (VEGF) and the VEGF receptor (VEGFR2) expression along with activation of the eNOS [41].

More recently, various clinical findings and animal experimentations have abundantly shown that RDN does improve the insulin sensitivity and endothelial function, even though the mechanism(s) involved remains unknown. When endothelial dysfunction in type 2 diabetes (T2D) in a rat model with insulin resistance was studied to explore the mechanisms and the effects of RDN, the RDN decreased the plasma and renal tissue norepinephrine levels significantly. In addition, the Von Willebrand (VW) factor levels were also decreased. Interestingly, the level of nitric oxide (NO) in plasma was also significantly increased after RDN. Side by side comparison with the T2D group and the sham group revealed that the endothelium-dependent and endothelium-independent diastolic function of the RDN group was also significantly improved. On further investigation, the relative expression levels of the molecules such as ACE2, Beclin1, LC3, and eNOS were found to be higher, but the levels of sequestome protein (also known as p62 protein) were decreased. It was noteworthy that RDN could activate the p-AMPK expression and thus inhibit the expression of p-mTOR factor. In parallel in vitro experiments involving the cell culture, factors such as ACE2 helped activate the p-AMPK and that, in turn, was sufficient to inhibit the p-mTOR molecule, thereby inducing the autophagy process [43].

It is known that renal denervation improves the vascular endothelial function, but it is unclear whether it also assists in the endocardial endothelial function during HFrEF phenotype [44]. From our study, it appears that the RDN intervention also improves the endocardial-endothelial dependent cardiac function as seen from our results that employed the cardiac LV ring responses to acetylcholine (ACH). The attenuation in endothelial dependent cardiac function was expected post AVF procedure; however, the RDN intervention was able to improve the cardiac endothelial function as depicted in Figure 3. our results were further supported by echocardiography (ECHO) measurements stemming from the RDN procedure towards maintaining the HFpEF phenotype from HFrEF one. Although the renal denervation procedure helps improve the cardiac fibrosis and hypertrophy, it is unclear whether it also does in the HFrEF phenotype [45,46]. The problem pertaining to HFpEF in the past 25 years has led to a heightened interest in clinical investigations aiming for understanding the pathogenesis of this devastating disease. It is hoped that employing phenotyping along with genotyping of patient populations may lead to further improvement in outcomes from future clinical trials, thus helping to diminish the overall health burden of HFpEF in the targeted population. This very approach might spur and help identify other patient populations within the broad term of the HFpEF phenotype for future planned studies. The major difference in existing international guidelines aimed at treatments and device-based procedures is essentially based upon LVEF functioning that elevates its ascertainment to management decisions in HF, although it is not needed for the diagnosis of HF itself. The previous categorization of the patient populations with HF was based upon LVEF functioning and has, in fact, because of clinical trials managed to acquire significant importance. In this context, design concerns regarding sample size and event rate considerations led most of the previous treatment outcome trials in HF to focus on those patients with significantly reduced LVEF function. Due to this, the randomized clinical trials’ evidence on which to base the treatment decisions for those patients with preserved the LVEF function is much more limited in scope and appeared to be less definitive. This all has been changing now since the patients with HFpEF phenotype are being considered as an unmet need in the field of cardiology. Further, testing of new treatment modalities vis a vis placebo rather than on known or proven therapeutics and device-based procedures have enhanced the need for conducting randomized trials in the HF patients [47,48].

It is known that myocardial infarction (MI) increases the risk HF and atrial fibrillation (AF) in patients. It has been hypothesized that RDN may be able to suppress the atrial remodeling development. During the elucidation of the mechanism(s) of RDN towards the suppression of AF in the HF model after MI, it was demonstrated that RDN was able to reverse the atrial electrical as well as structural remodeling of the tissue and, thereby, could suppress the AF in an ischemic HF model. The authors revealed the beneficial effect of RDN might be, in fact, related to the prevention of phosphoinositide 3-kinase/AKT/endothelial nitric oxide synthase signaling pathway downregulation [49]. In our work we show that the RDN procedure influences the cardiac hypertrophy and fibrosis that were obvious after the AVF procedure in the hearts, but RDN intervention did not have much significant effect on cardiac hypertrophy or fibrosis (Figure 5), thus suggesting the preservation of ejection fraction (EF) despite not influencing the cardiac hypertrophy and fibrosis processes. Although more work needs to be performed in validating our observations, based upon our results in the experimental mice, we do opine that RDN intervention could certainly assist in decreasing the activity of the sympathetic nerves (i.e., the renal sympathetic afferent nerve-hypothalamus-renal sympathetic efferent nerve circuit system) as schematized in Figure 6, and that could help improve the left ventricular governed heart function in patients who remain prone to failure and cardiac death.

## 4. Materials and Methods

### 4.1. Creation of the Aorta Vena Cava Fistula (AVF) in a Mouse Model

To create chronic/congestive (cardio-pulmonary) heart failure (CHF) in mice, an aorta-vena cava fistula (AVF) below the kidneys was performed, as mentioned previously, between the aorta and caudal vena cava approximately 0.5 cm below the kidneys using a 30-gauge sterile needle under aseptic surgical condition [50,51,52,53]. The sham surgery was used as control in a separate group of mice. Why does the AVF model of HFpEF to HFrEF work better? As mentioned earlier, there are basically four ways to create the experimental CHF procedure (*vide supra*). By creating the fistula, the red blood enters the vena cava without an injury to the heart. In fact, this model mimics the real CHF phenotype, such as the aging process, wherein we age and our preload keeps increasing beyond the level that our heart can help pump out, due to the weakened cardiomyocytes in the aging heart. This technique of creating the AVF also involves right ventricle to lung to left ventricle, thus enforcing the congestive heart failure (CHF) phenotype. After the AVF process, the heart transitions from the HFpEF to HFrEF over time. Again, our lab is one of the first labs that is instrumental in creating and perfecting the AVF procedure. To determine the protective role of renal nerve denervation (RDN), the RDN procedure was performed in one group of the AVF mice, and in a sham mouse group side by side. The mice were housed in the core animal care facility at the University of Louisville School of Medicine. To study the role(s) of renal nerve in volume sensing, experiments were carried out in renal “in-nerve and de-nerve” animals. The denervation procedure was performed by surgically removing all the nerves ending around renal artery and subsequently treated with phenol reagent, as described earlier [30,31,54]. These are novel experiments because the cardiac vascular reactivity in volume overload denervated condition has not been studied in detail before. 

### 4.2. Serial and Longitudinal Echocardiograms (ECHOs) in Mice 

To determine HFpEF and HFrEF phenotypes, serial and longitudinal ECHOs in mice at 1–6 weeks (HFpEF) and 12–16 weeks (HFrEF) were carried out. The concentric cardiac hypertrophy, a marker of HFpEF, was assessed via the long-axis ECHO as described earlier [53]. After the mice were sacrificed, the concentric hypertrophy was assessed by histology, as we have described previously [55]. CHF precedes chronic volume overload; therefore, the aorta-vena cava fistula (AVF) procedure was performed below the kidney in the male wild type (WT, C57BL/6J) mice with and without renal denervation (RDN) intervention. To differentiate the HFpEF vs. HFrEF, serial/longitudinal ECHOs were performed to determine the ejection fraction (EF) from 1 to 6 weeks after the AVF procedure by Vevo 2100 echocardiography equipment. The EF > 50% was denoted as HFpEF, and the EF < 50% was taken as HFrEF. The changes in the left ventricle (LV) filling volume, and the early diastolic velocity were measured. A pulse-wave pressure of >40 mmHg was considered as the chronic congestive cardiopulmonary heart failure (CHF) phenotype.

### 4.3. Measurement of Renal and Blood Exosomal Levels of EPO

In addition, to focus on the exosomes containing the erythropoietin (EPO) regulated vascular density and cardiac function in HFpEF versus HFrEF, the role(s) of the circulating exosomal EPO in the context of the expression levels of molecules such as eNOS, ACE1, ACE2, SGLT1/2, and nephrin were assessed. To carry out the assessment, the renal and cardiac EPO, eNOS, Wnt1, and β-catenin, and respective levels of the renal (cortical) and medullary expression amounts of the caspases were also studied and quantified in sham, and AVF at 4–6 weeks following the AVF procedure. Circulating exosomes containing the EPO were analyzed by Western blotting (WB) experiments. The fibrosis was assessed by trichrome (collagen) and van-Giessen (elastin) staining procedure. A quantitative collagen estimation was carried out biochemically by the help of hydroxy proline measurements. Glomerular function rate (GFR) elicits overall function; therefore, reactivity to acetylcholine (ACH) was measured via the ex vivo preparation as described earlier [56]. 

### 4.4. Statistical Analysis

Data obtained from the same samples were used to express results in terms of percentage change(s) relative to the controls (prior to treatment/procedure). Experiments using commercial biochemical assays were performed in triplicate. Statistical significance was determined by student’s *t*-test for two groups, or two-way ANOVA (analysis of variance) was used to compare the respective data with wild type, with RDN or without RDN according to the Bonferroni correction [57]. The differences were considered statistically significant at *p* < 0.05.

## Figures and Tables

**Figure 1 ijms-24-07302-f001:**
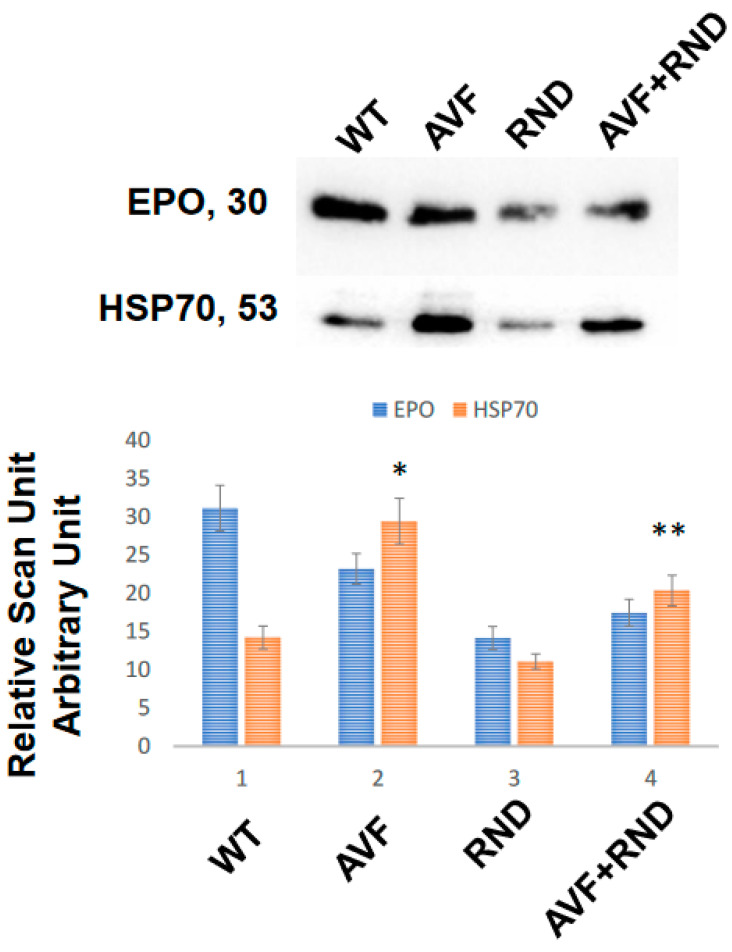
Blood levels of circulating exosomes in the C57BL/6J (WT-controls), AVF, RND, and AVF-RND male mice at 16 weeks. The upper gel panel represents levels of erythropoietin (EPO), and lower panel depicts exosomal marker heat shock protein 70 (HSP70). The bar graph represents mean ± SD from *n* = 5; *, *p* < 0.05 compared to WT; **, *p* < 0.05 compared with AVF.

**Figure 2 ijms-24-07302-f002:**
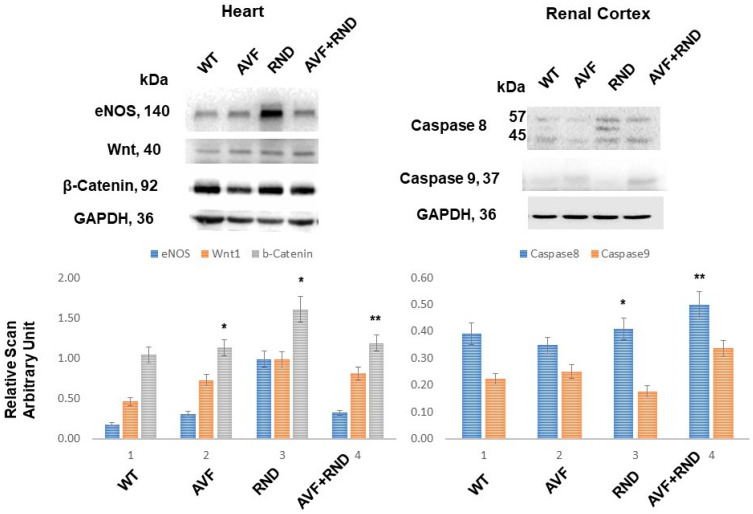
Representative Western blotting analyses of the cardiac tissue for endothelial nitric oxide synthase (eNOS), wingless-related integration (wnt1), and β-catenin signaling pathway molecules and from the renal-cortex tissues for caspase 8 and −9 in samples obtained from C57BL/6J WT-control mice, AVF, RND, and AVF-RND male mice at 16 weeks. The bar graphs represent the mean ± SD from *n* = 5; *, *p* < 0.05 compared to WT; **, *p* < 0.05 compared with AVF.

**Figure 3 ijms-24-07302-f003:**
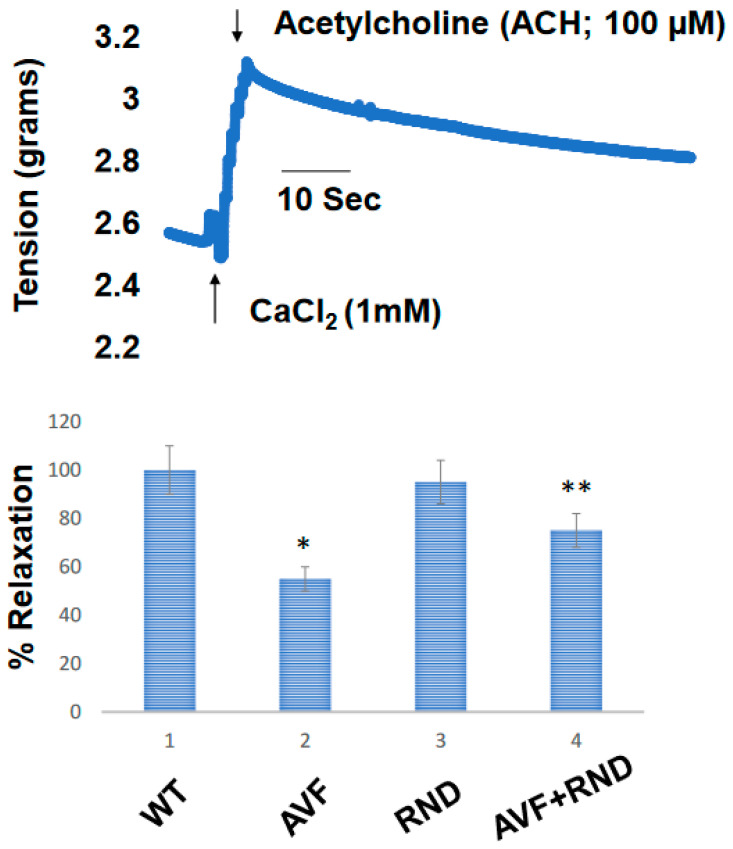
Cardiac endothelial dependent endocardial endothelial function measured in cardiac left ventricle (LV) ring prepared from C57BL/6J WT-control, AVF, RND, and AVF-RND male mice at 16 weeks. The rings were contracted with CaCl2 (1 mM), and acetylcholine was added to relax the cardiac rings. The bar graph represents the percent relaxation from CaCl_2_ (1 mM) contraction by 100 µM acetylcholine (ACH). The mean ± SD from *n* = 5; *, *p* < 0.05 compared to WT; **, *p* < 0.05 compared with AVF.

**Figure 4 ijms-24-07302-f004:**
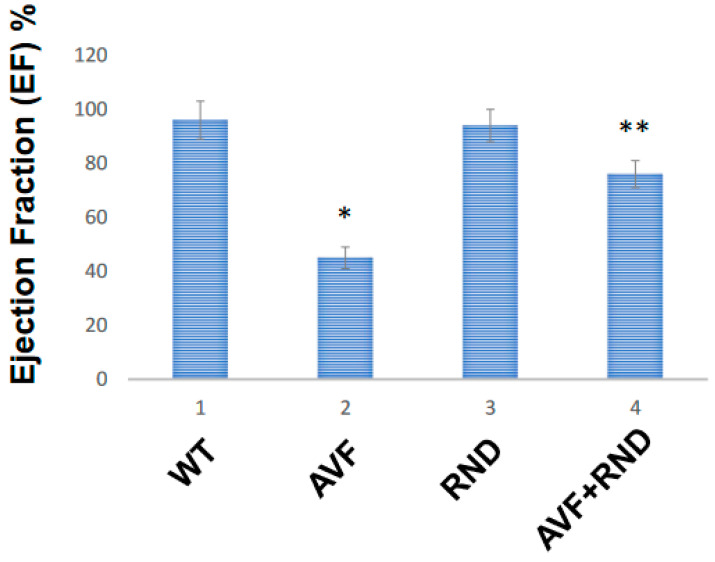
The cardiac ECHO data from C57BL/6J WT-control, AVF, RND, and AVF-RND male mice at 16 weeks. The bar graph represents percent (%) ejection fraction of the male mice hearts. The mean ± SD from *n* = 5; *, *p* < 0.05 compared to WT; **, *p* < 0.05 compared with AVF.

**Figure 5 ijms-24-07302-f005:**
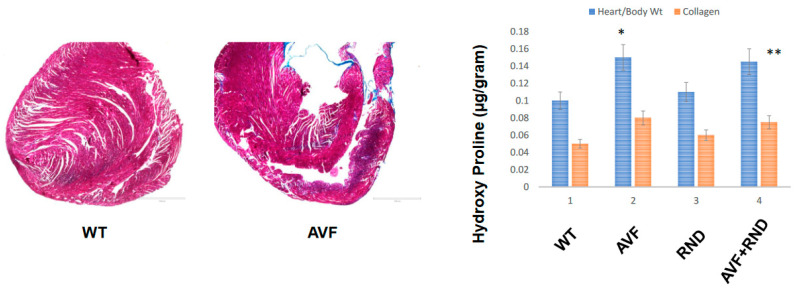
Left panel highlights the collagen trichrome-blue staining for the C57BL/6J WT and AVF male mice at 16 weeks, while the right panel indicates cardiac hypertrophy as estimated by heart weight/body weight ratios and collagen fibrosis via hydroxy proline measurements in C57BL/6J WT-control, AVF, RND, and AVF-RND male mice at 16 weeks. The bar graph represents hypertrophy in ratio (g/g) and hydroxyproline (µg/g of the tissue). The mean ± SD from *n* = 5; *, *p* < 0.05 compared to WT; **, *p* < 0.05 compared with AVF, the scale bar in the left panel = 1000 µM.

**Figure 6 ijms-24-07302-f006:**
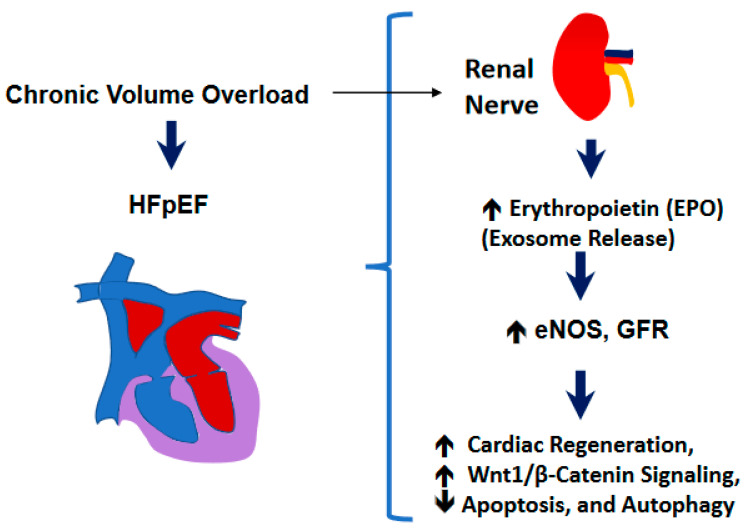
A schematic showing the renal mechanism of cardiac preserved ejection fraction (EF) during heart failure (HF).

## Data Availability

The research data will be made available on request as per the Institution’s policy.

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
