# Peer review of "Renal Denervation Helps Preserve the Ejection Fraction by Preserving Endocardial-Endothelial Function during Heart Failure"

_ijms, 2023, doi:10.3390/ijms24087302_

Round 1

Reviewer 1 Report

This work discussed RDN procedure influences the cardiac hypertrophy and fibrosis that were obvious after the AVF procedure in the hearts, but RDN intervention did not have many significant effects on cardiac hypertrophy and fibrosis. RDN could help improve the left ventricular governed heart function in patients who remain prone to failure, and cardiac death. The author used AVN model to explain the impact of RDN on fibrosis and apoptosis. Caspase 8 is elevated to protect the heart in RDN model. The work also provided a clear cartoon to explain the mechanism.

Author Response

Dear Reviewer #1:

We thank, and appreciate you for your time, and the positive comments about our work. You are correct that the RDN procedure does influence the cardiac hypertrophy, and fibrosis that were clear after the AVF procedure. You are also supportive that RDN could, in fact, help improve the left ventricular cardiac function in the patients who often remain prone to heart failure, and cardiac death.

Further, the schematic/cartoon is truly helpful, and explain the overall mechanism that most likely takes place, in a real life situation. Again, we remain thankful to you for the encouragement.

Reviewer 2 Report

This paper is an important contribution in this area. The study is well written and interesting. Methods, results and discussion are clearly presented. References are fine. I wonder if Authors have measured Endothelin 1 levels. This could be helpful in predicting endothelial function and left ventricle function outcomes following denervation. 

Author Response

Dear Reviewer #2:

We do appreciate your positive comment(s) about our work. You are absolutely correct that our paper is an important contribution in this area.  Further, as you have noticed that our study is well-designed, and also the paper is nicely written.

The methods, results, and discussion parts are clearly presented in this paper, as you might have noticed. Although, the references are fine but we have tried further to update them in light of the importance of this particular disease phenotype.

Regarding the measurement of "Endothelin 1" levels, we wish we could; however, we did carry out the function assessment of the "Endocardial- Endothelial" function using the Cardiac Left Ventricle (LV) Rings since we opine that this functional assessment is also helpful in predicting the endothelial function, and the left ventricle function outcomes following the denervation process. 

Again, we thank the reviewer for the valuable time, and helpful comments.